# Associations with HIV preexposure prophylaxis use by cisgender female sex workers in two Ugandan cities

**Richard Muhindo**[1,2*], **Rachel King**[3], **Whitney Irie**[4], **Andrew Mujugira**[2],
**Edith Nakku-Joloba**[5], **Stephen Okoboi**[2], **Patience Muwanguzi**[1], **Eva Laker Odongpiny**[2],
**Nazarius Mbona Tumwesigye**[5], **Barbara Castelnuovo**[2]

1 Department of Nursing, Makerere University College of Health Sciences, Kampala, Uganda, 2 Infectious Diseases Institute, Makerere University College of Health Sciences, Kampala, Uganda, 3 University of California, San Francisco, California, United States of America, 4 Boston College, Chestnut Hill, Massachusetts, United States of America, 5 Makerere University School of Public Health, Kampala, Uganda

☺ These authors also contributed equally to this work
* r.muhindo@yahoo.com

## Abstract

### Background

Sex workers of all genders have a high risk of HIV acquisition and are a priority population for HIV pre-exposure prophylaxis (PrEP). We aimed to assess current oral PrEP use and associated factors among cisgender female sex workers (FSW) in two Ugandan cities.

### Methods

We administered a survey questionnaire to 236 HIV-negative FSW in the cities of Mbale and Mbarara from January to March 2020. The survey was nested in a quasi-experimental study to assess the effect of peer education and text message reminders on the uptake of regular sexually transmitted infection (STI) and HIV testing. Using interviewer-administered questionnaires, we obtained data on current self-reported tenofovir-based oral PrEP use. We used modified Poisson regression with robust standard errors to evaluate the factors associated with current oral PrEP usage.

### Results

Nearly 70% of FSWs reported taking an HIV test during the past three months. Among the respondents, 33% (33/100) in Mbale and 67% (91/136) in Mbarara reported having ever heard of PrEP. However, only 9.7% (23/236) self-reported currently taking oral-PrEP. In Mbarara, FSWs were twice as likely to be aware of or use oral PrEP than those in Mbale (adjusted prevalence ratio [aPR] 2.33; 95% confidence interval (CI) 1.19–3.97; p = 0.01). Additionally, current use was positively associated with attainment of secondary (aPR 2.50; 95% CI: 1.14–5.45; p = 0.02) or tertiary education (aPR 3.12; 95% CI: 1.09–8.96; p = 0.03).

**Data availability statement:** The datasets used during the current study are available as Supporting Information.

**Funding:** This study was supported through grant number D43TW010132 supported by the Office Of The Director, National Institutes Of Health (OD), National Institute Of Dental & Craniofacial Research (NIDCR), National Institute Of Neurological Disorders And Stroke (NINDS), National Heart, Lung, And Blood Institute (NHLBI), Fogarty International Center (FIC), National Institute On Minority Health And Health Disparities (NIMHD) and a grant from the Ugandan Academy of Health Innovation and Impact. The Ugandan Academy is initially funded by Janssen, the Pharmaceutical Companies of Johnson & Johnson, as part of its commitment to global public health through collaboration with the Johnson & Johnson Corporate Citizenship Trust. However, the funders had no role in study design, data collection and analysis, publication decisions, or manuscript preparation.

**Competing interests:** The author(s) declare that they have no competing interests.

**Abbreviations:** CI: confidence interval; FSW: female sex workers; HIV: human immunodeficiency virus; IQR: interquartile range; PrEP: pre-exposure prophylaxis; STI: sexually transmitted infection; WHO: World Health Organization.

## Conclusion

PrEP use in this cohort of FSWs was low and was associated with location and level of education. To increase PrEP uptake among FSWs, targeted educational campaigns and implementation studies are needed, particularly for those with lower levels of education.

## Introduction

HIV pre-exposure prophylaxis (PrEP) has become a crucial component of global HIV prevention strategies, particularly for female sex workers (FSWs) and other populations at increased risk for HIV acquisition [1–3]. In 2022, the relative risk of acquiring HIV was nine times higher for sex workers than in the wider adult (aged 15–49 years) population globally [4]. Daily oral tenofovir and co-formulated emtricitabine-tenofovir disoproxil fumarate (FTC/TDF) have been shown to significantly reduce the incidence of HIV [5]. Specifically, studies conducted among heterosexual men and women in Uganda and Kenya demonstrated a 67% reduction in HIV incidence with daily oral tenofovir, and a 75% reduction with co-formulated FTC/TDF [5]. The implementation and adherence to PrEP among FSWs, holds promise in the quest to end the HIV epidemic, particularly in sub-Saharan Africa (SSA), the region most heavily affected by HIV [6–8]. In 2022, sex workers accounted for an estimated 5.2-15% of new HIV infections in SSA [9]. Importantly, FSWs act as a bridge population - up to 15% of HIV infections in the general adult female population are attributable to sex work [3]. In Uganda, FSWs exhibit a high HIV prevalence, ranging from 31% to 52%, and account for 7-11% of new HIV infections [10,11].

Studies across SSA have reported high acceptability of PrEP among FSWs, ranging from 53% to 98% across different regions [12–15]. Surveys in Nigeria, Tanzania, and Uganda reported high willingness to use PrEP among FSWs, with 91% in Nigeria, 98% in Tanzania, and 91% in Uganda expressing interest [12,14,15]. In contrast, a survey in Ghana reported a lower willingness to use PrEP among FSWs, with only 53.9% expressing interest [13].

In Uganda, oral PrEP roll-out among HIV-negative persons at increased risk of HIV acquisition, including FSW, started in July 2017 [16]. While PrEP acceptability among FSWs in Uganda and SSA has been well-documented, data on PrEP use outside demonstration projects remains limited. Most studies on PrEP have focused on initiation, with many conducted within the context of PrEP demonstration projects. These projects aim to introduce PrEP in real-world settings, assess its feasibility and effectiveness, and identify implementation challenges [14,17–19]. However, there is a need for more research on PrEP use among the general FSW population. This study aimed to assess current oral PrEP use and associated factors among FSWs in two Ugandan cities.

## Methods

### Study design and setting

This cross-sectional survey conducted between January and March 2020 was nested in a quasi-experimental study that assessed the effect of peer education and text message reminders on the uptake of regular sexually transmitted infection (STI) and HIV testing among sex workers in Uganda, as previously described [20]. The survey questionnaire was administered to 236 HIV-negative FSWs in Mbale and Mbarara cities in Uganda's Eastern and Western regions, respectively. Mbarara and Mbale served as the intervention and control sites, respectively, for our quasi-experimental study [20], and were purposely selected for this survey.

## Sampling and recruitment procedures

The study sample consisted of women ≥ 18 years old who self-reported selling sex for goods or money for at least 6 months and not living with HIV. Prior to participant recruitment, a mapping exercise was conducted to gather insights into the dynamics of sex work in each city. In consultation with local health workers and barmaids, high volume sex work hotspots (locations frequented by sex workers or where sex work occurs) were enlisted. In Uganda, barmaids may engage in sex work or facilitate connections between sex workers and clients [21] The mapping exercise revealed three primary typologies of sex work venues: street-based, lodge-based, and bar/club-based, informing the development of our recruitment strategy. Sex work hotspots constituted the primary sampling units. Respondents were recruited from sex work hotspots in Mbarara and Mbale, two urban areas with a high prevalence of sex work [22]. Through peer referrals we enrolled study respondents as previously described [20,23].

## Data collection

Data were collected through an interviewer-administered questionnaire. Interviews were conducted by a trained team member fluent in the respondent's preferred language (English, Luganda or Runyankore). Questionnaire domains included demographic characteristics, condom use and STI/HIV testing behaviors, oral PrEP awareness (defined as whether the FSW knew about oral PrEP pills taken daily by an HIV-negative person to prevent HIV acquisition), and current use (defined as self-reported use of daily oral pills to prevent HIV acquisition at the time of the interview).

## Statistical analysis

The primary outcome was current PrEP use, defined as self-reported use of daily oral pills to prevent HIV acquisition at the time of the interview. Demographic characteristics, condom use, syphilis and HIV testing behaviors, and PrEP use were described using proportions. A modified Poisson regression with robust standard errors was used to determine factors associated with PrEP use [24,25]. Both deviance and Pearson chi-square goodness of fit tests were conducted to assess the model. Crude and adjusted prevalence ratios (PR) and their 95% confidence intervals (CI) were estimated. A two-sided p-value of 0.05 or less was considered statistically significant. All statistical analyses were performed using Stata version 15.0 software (StataCorp, College Station, TX).

## Ethical clearance

The Higher Degrees Research and Ethics Committee of Makerere University School of Public Health and the Uganda National Council for Science and Technology (HS 2403) granted ethics approval for this study. All respondents provided written informed consent in English or their local language, and were compensated for their time.

## Results

A total of 236 FSW were included in the analysis, with a median age of 27 years (interquartile range [IQR] 23-31) in Mbale and 26 years (IQR 22-30) in Mbarara. Their demographic characteristics were broadly similar between the two cities, except for the average number of clients per week and marital status (**Table 1**). The median duration in sex work was respectively 38 months in Mbale and 36 months in Mbarara. FSWs in Mbale reported a significantly higher number of clients per week, with a median of 25 compared to 15 in Mbarara.

Table 1. Characteristics of FSW enrolled in the survey stratified by city.

| Variable | Mbale (n = 100) | Mbarara (n = 136) | p-value |
|---|---|---|---|
| Age in years, median (IQR) | 27 (23–31) | 26 (22–30) | 0.73 |
| Duration in months in sex work, median (IQR) | 38 (24–70) | 36 (18–66) | 0.35 |
| Clients per week, median (IQR) | 25 (20–31) | 15 (8–35) | **0.01*** |
| Education, n (%) | | | 0.2 |
| None | 15 (15.0) | 20 (14.7) | |
| Primary | 55 (55.0) | 67 (49.3) | |
| Secondary | 30 (30.0) | 45 (33.1) | |
| Tertiary | 0 | 4 (2.9) | |
| Marital status, n (%) | | | **0.01*** |
| Single | 61 (61.0) | 57 (42.0) | |
| Separated | 31 (31.0) | 74 (54.4) | |
| Widow | 6 (6.0) | 1 (0.7) | |
| Married | 2 (2.0) | 4 (2.9) | |
| Sex work venue, n (%) | | | N/A |
| Street | 87 (43.0) | 52 (25.0) | |
| Home | 1 (0.5) | 2 (1.0) | |
| Lodge | 84 (42.0) | 62 (30.0) | |
| Bar/club | 24 (12.0) | 89 (43.0) | |

**FWS**, female sex worker; **IQR,** interquartile range;

*statistically significant

In both cities, over three quarters of the FSW had obtained either primary or secondary level of education, with 85% in Mbale and 82% in Mbarara having attained this level. Marital status significantly differed between the cities, with most FSWs in Mbale identifying as single (61%) and most in Mbarara describing themselves as separated (54%). The main venues for soliciting clients were similar, with streets, lodges, and bars/clubs being the most common in both cities.

Significant variations were observed among FSWs in Mbale and Mbarara regarding condom use, syphilis and HIV testing, PEP and PrEP awareness and usage (p < 0.001 for all comparisons) (**Table 2**). Compared to Mbale, FSWs in Mbarara reported higher rates of recent syphilis and HIV testing. Lifetime syphilis testing was significantly higher in Mbarara (90%) than in Mbale (61%), with recent testing (within the past three months) also more common in Mbarara (81%) than in Mbale (26%). Although lifetime HIV testing rates were similar between the two sites (Mbarara: 97%, Mbale: 93%), recent HIV testing was more frequent in Mbarara (82%) than in Mbale (52%).

Awareness and use of PrEP remained low in both cities; however, these percentages were higher in Mbarara with 67% awareness, 23% ever using PrEP, and 17% currently using it compared to Mbale with only 33%, 9%, and no current users respectively. Also, PEP awareness and use were lower in Mbale (awareness: 45%, use: 8%) compared to Mbarara (awareness: 66%, use: 15%).

After adjusting for age, marital status, clients per week, and condom use frequency in multivariate analysis, residing in Mbarara (adjusted prevalence ratio [aPR] 2.33; 95% confidence interval (CI) 1.19–3.97; p = 0.01) and attaining secondary education (aPR 2.50; 95% CI: 1.14–5.45; p = 0.02), or tertiary education (aPR 3.12; 95% CI: 1.09–8.96; p = 0.03) were associated with current PrEP use (**Table 3**).

Table 2. Condom use, STI/HIV testing behaviours, and PrEP use stratified by city.

| Variable | Mbale (N = 100) Proportion (95% CI) | Mbarara (N = 136) Proportion (95% CI) | p-value |
|---|---|---|---|
| **Condom use at last sexual activity** | | | **0.001**[*] |
| Yes | 0.91 (.83–.95) | 0.73 (.65–.79) | |
| **Condom use practice** | | | **0.001**[*] |
| Always | 0.67 (.57–.75) | 0.31 (.24–.39) | |
| Sometimes | 0.33 (.24–.43) | 0.69 (.61–.76) | |
| **Syphilis testing history** | | | **0.001**[*] |
| Ever tested | 0.61 (.51–.70) | 0.90 (.84–.94) | |
| Tested in the prior 3 months | 0.26 (.16–.39) | 0.81 (.73–.87) | |
| **HIV testing history** | | | |
| Ever tested | 0.93 (.85–.96) | 0.97 (.92–.98) | 0.14 |
| Tested in the prior 3 months | 0.52 (.41–.62) | 0.82 (.74–.88) | **0.001**[*] |
| **PEP awareness and use** | | | **0.001**[*] |
| Ever heard of PEP | 0.45 (.36–.63) | 0.66 (.57–.75) | |
| Ever used PEP | 0.08 (.03–.12) | 0.15 (.09–.23) | |
| **PrEP awareness and use** | | | **0.001**[*] |
| Ever heard | 0.33 (.21–.48) | 0.67 (.58–.76) | |
| Ever used oral PrEP | 0.09 (.01–.11) | 0.23 (.16–.26) | |
| Currently using oral PrEP | 0 (0.0) | 0.17 (.11–.25) | |

**CI**, confidence interval; **95% CI**, 95% confidence interval;

[*] statistically significant

## Discussion

This cross-sectional study investigated current oral PrEP use among FSWs in two Ugandan cities. Less than half (42.3%) of the FSWs reported being aware of oral PrEP, with only 9.7% reporting current use. However, awareness and use significantly varied by location. Notably, FSWs who had attained secondary or tertiary education were 2.5-3 times more likely to report current use, consistent with previous studies conducted in SSA [26–28]. These studies have consistently shown a positive correlation between education level and PrEP usage, highlighting the need for implementation research on strategies to promote PrEP use among FSWs with lower education levels.

Our finding that FSWs in Mbarara were twice as likely to be aware of PrEP and report current use compared to those in Mbale highlights limited awareness and uptake in certain Ugandan urban locations. This disparity may be attributed to Mbarara's high adult HIV prevalence (14.4% vs 5.4% in Mbale) [10], which has likely fueled ongoing HIV-focused research and increased community-level awareness of HIV prevention [29,30]. Furthermore, the presence of a PrEP demonstration project targeting women at increased risk of HIV acquisition in Mbarara may also contribute to the observed difference [30]. Targeted educational campaigns to FSWs regarding the benefit and availability of oral PrEP services are still needed.

Regional variations in PrEP usage have been observed among FSWs in East Africa. While our finding of low current PrEP use (9.7%) is comparable to a Kenyan study (24%) [26], a recent Tanzanian study reported notably higher use (97%) [27]. These disparities likely stem from contextual differences in service organization, provider characteristics, and recipient factors. Existing literature cites various barriers to non-oral PrEP use among FSWs, including

**Table 3. Factors associated with PrEP use among FSW.**

| Variable | Crude prevalence ratio (95% CI) | p-value | Adjusted prevalence ratio (95% CI) | p-value |
|---|---|---|---|---|
| **City** | | | | |
| Mbale | Reference | | | |
| Mbarara | 2.41 (1.23–4.73) | **0.01**[*] | 2.33 (1.19–3.97) | **0.01**[*] |
| **Age, years** | | | | |
| ≤ 24 | Reference | | | |
| 25–29 | 0.63 (.27–1.5) | 0.29 | 0.69 (.29–1.64) | 0.41 |
| 30+ | 0.74 (.34–1.6) | 0.45 | 0.97 (.45–2.09) | 0.94 |
| **Education** | | | | |
| None or primary | Reference | | | |
| Secondary | 2.08 (.99–4.34) | 0.05 | 2.50 (1.14–5.45) | **0.02**[*] |
| Tertiary | 2.32 (.88–6.14) | 0.08 | 3.12 (1.09–8.96) | **0.03**[*] |
| **Marital status** | | | | |
| Married | Reference | | | |
| Single | 1.62 (1.20–2.7) | **0.02**[*] | 1.35 (.67–2.98) | 0.09 |
| Has a boyfriend | 1.32 (1.09–1.9) | **0.03**[*] | 0.92 (.45–2.10) | 0.21 |
| **Clients per week** | | | | |
| ≤10 | Reference | | | |
| 11–20 | 1.16 (.52–2.59) | 0.74 | 1.45 (.62–3.38) | 0.38 |
| 21+ | 0.65 (.28–1.46) | 0.81 | 0.72 (.30–1.73) | 0.47 |
| **Condom use practice** | | | | |
| Always | Reference | | | |
| Sometimes | 2.37 (1.04–5.42) | **0.04**[*] | 1.25 (.61–2.53) | 0.53 |

**CI**, confidence interval; **95% CI**, 95% confidence interval;

[*] statistically significant

limited access, lack of awareness, negative perceptions, stigma, and pill burden [26,31,32]. Despite these insights, targeted interventions are still necessary to improve PrEP uptake and usage.

This cross-sectional study, conducted prior to the first COVID-19 lockdown in Uganda, provides a snapshot of PrEP use among FSWs in Mbale and Mbarara. However, the findings may not capture trends in PrEP use over time. Additionally, self-reported PrEP use may be subject to social desirability bias. Moreover, the study's sample was limited to two cities and may not be generalizable to all FSW communities in Uganda. Furthermore, peer referrals, a non-random sampling strategy was used to enroll study respondents. Despite these limitations, the results highlight the need for targeted educational campaigns and implementation studies to increase PrEP uptake among FSWs, particularly in areas with low awareness and usage beyond demonstration projects.

## Conclusions

The low PrEP usage in this FSW cohort underscores the need for implementation studies that evaluate strategies to improve uptake, particularly among those with lower levels of education.

## Supporting information

**S1 File. Data set for the main analysis.**
(DTA)

## Acknowledgments

Richard Muhindo is a Fogarty UCGHI GloCal Fellow supported by the Fogarty International Center of the National Institutes of Health (NIH) and the University of California Global Health Institute (UCGHI). We extend our sincere gratitude to the leadership of the cities of Mbale and Mbarara for their support and facilitation of the data collection process. Additionally, we acknowledge the contributions of Collins Twesigye and Jonan Mweteise, who served as research assistants.

## Author contributions

**Conceptualization:** Richard Muhindo.

**Data curation:** Richard Muhindo.

**Formal analysis:** Richard Muhindo, Andrew Mujugira, Nazarius Mbona Tumwesigye.

**Investigation:** Richard Muhindo.

**Methodology:** Barbara Castelnuovo.

**Supervision:** Barbara Castelnuovo.

**Validation:** Rachel King, Nazarius Mbona Tumwesigye.

**Visualization:** Rachel King, Andrew Mujugira.

**Writing – original draft:** Richard Muhindo.

**Writing – review & editing:** Rachel King, Whitney Irie, Andrew Mujugira, Edith Nakku-Joloba, Stephen Okoboi, Patience Muwanguzi, Eva Laker Odongpiny, Nazarius Mbona Tumwesigye, Barbara Castelnuovo.

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
