## [Decision Letter · Decision Letter 0]

4 Dec 2024

PONE-D-24-24744Associations with HIV Preexposure Prophylaxis Use by Cisgender Female Sex Workers in Two Ugandan CitiesPLOS ONE

Dear Dr. Muhindo,

Thank you for submitting your manuscript to PLOS ONE. After careful consideration, we feel that it has merit but does not fully meet PLOS ONE’s publication criteria as it currently stands. Therefore, we invite you to submit a revised version of the manuscript that addresses the points raised during the review process.

We look forward to receiving your revised manuscript.

Kind regards,

Hamufare Dumisani Mugauri, Ph.D. Medicine and Health Sciences

Academic Editor

PLOS ONE

2. In the online submission form, you indicated that [The datasets used during the current study are available from the corresponding author on request]. All PLOS journals now require all data underlying the findings described in their manuscript to be freely available to other researchers, either 1. In a public repository, 2. Within the manuscript itself, or 3. Uploaded as supplementary information. This policy applies to all data except where public deposition would breach compliance with the protocol approved by your research ethics board. If your data cannot be made publicly available for ethical or legal reasons (e.g., public availability would compromise patient privacy), please explain your reasons on resubmission and your exemption request will be escalated for approval.

Additional Editor Comments (if provided):

Reviewers' comments:

Reviewer's Responses to Questions

**Comments to the Author**

1. Is the manuscript technically sound, and do the data support the conclusions?

Reviewer #1: Yes

Reviewer #2: Yes

2. Has the statistical analysis been performed appropriately and rigorously? 

Reviewer #1: Yes

Reviewer #2: I Don't Know

3. Have the authors made all data underlying the findings in their manuscript fully available?

Reviewer #1: Yes

Reviewer #2: Yes

4. Is the manuscript presented in an intelligible fashion and written in standard English?

Reviewer #1: Yes

Reviewer #2: Yes

5. Review Comments to the Author

Reviewer #1: Dear authors,

Thank you for submitting this manuscript. Here are my comments:

- Within the first paragraph of the introduction, please be more specific with terms such as "successful uptake" and "highly effective solution." Please provide greater background on PrEP, its efficacy, and context within Uganda.

- Please describe why Mbale and Mbarara were selected for the research project within the introduction.

- Please describe the parameters used when conducting the mapping exercise. What qualifies as a "sex work hot spot?"

- Please further describe the impact of this research within the discussion and conclusions.

Reviewer #2: This is an interesting and clearly written manuscript that describes PrEP use and associated factors among female sex workers in two Ugandan cities.

I believe the authors could take several steps to improve the manuscript:

Introduction:

Line 3: Avoid use of the word “solution,” which implies 100% effectiveness.

Lines 3, 11: Avoid use of the term “high-risk;” instead, specify “population at increased risk for HIV infection” (see CDC guidance on preferred terminology: https://www.cdc.gov/health-communication/php/toolkit/preferred-terms.html).

Lines 4-6: The authors should focus on FSWs, rather than “high-risk populations” as a whole, and then provide greater context regarding the role of FSWs in the HIV epidemic in sub-Saharan Africa (i.e., beyond stating that they are one of many “high-risk” populations).

Lines 7-8: This is misleading. A number of other populations at increased risk for HIV (e.g., long-distance truck drivers and people who inject drugs, among others) were included in the meta-analysis in the systematic review cited.

Lines 9-10: References 7 and 8 evidence PrEP acceptability rates of 91% and >50%, respectively; it is not clear what data supports the 100% acceptability rate stated here.

Lines 13-19: The authors should define “PrEP persistence” and clarify the distinction made, if intentional, between PrEP persistence and “long-term use” in lines 18-19. They should also clarify what is meant by “diverse settings” on line 19 and why this is a particular need.

Line 21: The authors should clarify what is meant by “everyday practice.”

Methods:

Line 15: The authors should clarify the language(s) in which the questionnaires were administered and the proficiency of both parties (interviewer and participant) in the language(s).

Results:

Lines 15-17: “Average number of clients per week” may be better described as a behavioral factor than a “demographic characteristic.”

Line 21: Based on the data in Table 1, the percentage of FSWs with a primary or secondary level of education in Mbale is 90% not 85%.

Line 22: The authors should clarify that marital status was a significant difference between the two cities.

Lines 4-6 (page 7): The authors should include PEP awareness and usage in the list of significant variations here. They should also consider specifying “syphilis testing (ever and prior three months)” and “HIV testing (prior three months only)” rather than generalizing as “STI/HIV testing.”

Discussion:

Lines 17-18 (page 7): The authors should opt for more precise language than “generally low” to describe what are the principal findings of the study. Additionally, in the abstract, the authors describe PrEP use in this cohort to be “very low,” which is substantially different from how it is described here. The authors should be consistent with the language used to describe these findings.

Lines 21-22 (page 7): The authors should specify the “services” to which they refer here.

Lines 22-4 (pp. 7-8): The authors should switch the order of these sentences (i.e., “Notably…” before “The differences in…”) to present the relevant findings prior to the speculative clause.

Lines 2-3 (page 8): The authors should provide a reference here for the PrEP demonstration project in Mbarara that is mentioned.

Lines 4-5: The authors should refine this sentence (“Despite…”) to be more precise. This study assessed current usage of oral PrEP services among FSWs in two cities in Uganda. The findings, as written here, seem to refer to the current usage of oral PrEP services among the general population across all of Uganda.

Lines 5-6: This sentence needs to be reworked. The findings that current usage of oral PrEP services among FSWs was low do not directly “contrast” those of previous studies that evidence high willingness among FSWs to use oral PrEP; rather, they simply highlight that willingness to use (which is different from “intention,” and would make this change as well) and actual usage are two different things.

Lines 11-16: The authors should specify what is meant by “key populations” on lines 15-16 i.e., populations at increased risk for HIV? Additionally, the authors should consider including a reference or two that are specific to the population of interest (i.e., FSWs) if possible.

Lines 17-18: Again, the authors should consider specifying what is meant by “those who need it most” i.e., populations at increased risk for HIV.

Lines 19-3 (pp. 8-9): The authors should include non-random sampling/ recruitment of survey respondents as a possible limitation to the study.

Conclusions:

The authors make sound conclusions and suggest practical next steps that follow from the data presented.

Table 2:

The authors should consider unitalicizing the values in this table for uniformity with Tables 1 and 3. They should also consider adding n values for the individual variables listed.

6. PLOS authors have the option to publish the peer review history of their article (what does this mean? ). If published, this will include your full peer review and any attached files.

**Do you want your identity to be public for this peer review?** For information about this choice, including consent withdrawal, please see our Privacy Policy .

Reviewer #1: **Yes: ** Elizabeth Anna Banyas

Reviewer #2: No

---

## [Author Response · Author response to Decision Letter 0]

17 Dec 2024

Dr. Hamufare D Mugauri

Academic Editor, Plos One

December 16th, 2024

Dear Dr. Hamufare,

Regarding PONE-D-24-24744

We are very grateful for the reviewer comments provided by the reviewers of our manuscript: “Associations with HIV Preexposure Prophylaxis Use by Cisgender Female Sex Workers in Two Ugandan Cities ". Reviewer comments are encouraging, and the reviewers appear to share our view that study findings are of public health importance. The suggestions offered by the reviewers have been greatly helpful. Please find below our point-by-point responses in italics, with new text in bold font:

Reviewer Comments

Reviewer 1

1. Within the first paragraph of the introduction, please be more specific with terms such as "successful uptake" and "highly effective solution." Please provide greater background on PrEP, its efficacy, and context within Uganda.

Response: We thank the reviewers for this observation. We apologize for the choice of words like successful, and highly effective solution that portrayed PrEP to be 100% effective or a solution to HIV acquisition or transmission, but also not providing a greater background on PrEP. We have now dropped the words, successful uptake, and highly effective solution. In addition, we provided some background context. The following amendment has been included in the background to address the concerns raised (page 4, lines 3-10).

In less than a decade, HIV pre-exposure prophylaxis (PrEP) has become a crucial component of global HIV prevention strategies, particularly for female sex workers (FSWs) and other populations at increased risk for HIV acquisition. In 2022, the relative risk of acquiring HIV was nine times higher for sex workers than in the wider adult (aged 15–49 years) population globally. Daily oral tenofovir and co-formulated emtricitabine-tenofovir disoproxil fumarate (FTC/TDF) have been shown to significantly reduce the incidence of HIV. Specifically, studies conducted among heterosexual men and women in Uganda and Kenya demonstrated a 67% reduction in HIV incidence with daily oral tenofovir, and a 75% reduction with co-formulated FTC/TDF

2. Please describe why Mbale and Mbarara were selected for the research project within the introduction.

Response: We thank the reviewer for this observation. We are sorry that in our submission we were unclear of why Mbale and Mbarara were selected. This sub-analysis was nested in a larger a quasi-experimental study that assessed the effect of peer education and text message reminders on the uptake of regular sexually transmitted infection (STI) and HIV testing among sex workers in Uganda. In the main study, Mbarara was the intervention site (peer education and short text mobile reminder messages), while Mbale was the control site. Prior to the quasi-experiment, FSWs in Mbarara and Mbale had low baseline syphilis and HIV testing intentions and practices between July and October 2018. We are unable to describe this in the background but rather in the methods section under study design and setting as we collected this data within a main study.

3. Please describe the parameters used conducting the mapping exercise. What qualifies as a sex work hotspot?

Response: We thank the reviewers for this observation. We apologize for not being clear. Otherwise, in our work a sex work hotspot meant locations frequented by sex workers or where sex work occurs. We have under the section on sampling and recruitment procedures described how the mapping was done, and what constituted a sex work hotspot. The text below has been added (page 6, lines 8-12);

A two-stage sampling design was used to recruit respondents. Prior to participant recruitment, a mapping exercise was conducted to gather insights into the dynamics of sex work in each city. In consultation with local health workers and barmaids, high volume sex work hotspots (locations frequented by sex workers or where sex work occurs) were enlisted.

4. Please further describe the impact of this research within the discussion and conclusions.

Response: We thank the reviewers for this recommendation. We are sorry our discussion was unable to be exhaustive. We have strengthened the discussion to include that the findings call for the need for awareness campaigns and implementation science studies. In particular the need to tailor the campaigns and studies to FSWs with low education levels.

Reviewer II

General comment

This is an interesting and clearly written manuscript that describes PrEP use and associated factors among FSWs in two Ugandan cities

Response: We thank the reviewers for this uplifting and encouraging comment.

Introduction:

1. Avoid use of the word “solution”, which implies 100% effectiveness

Response: We apologize the use of the term solution. We have now dropped it altogether from the current submission.

2. Avoid use of the term “high-risk”, instead specify population at increased risk for HIV infection

Response: We thank the reviewer for this observation and suggestion. We have adopted the suggestion of specifying the population or use the term population at increased risk for HIV infection throughout the manuscript.

3. The authors should focus on FSWs, rather than high-risk population as a whole and then provide greater context regarding the role of FSWs in the HIV epidemic in SSA

Response: We thank the reviewer for this guidance. We have now updated this section and provided the contribution for FSWs to HIV epidemic. The following text has been added (page 4, lines 13-18);

In 2022, an estimated 25% of new HIV infections in SSA were attributed to sex workers, their clients, and other populations at increased risk for HIV acquisition. Importantly, FSWs act as a bridge population - up to 15% of HIV infections in the general adult female population are attributable to sex work. In Uganda, FSWs exhibit a high HIV prevalence, ranging from 31% to 52%, and account for 7-11% of new HIV infections.

4. Lines 7-8: This is misleading. A number of other populations at increased risk for HIV were included in the metanalysis.

Response: We apologize for having cited this meta-analysis without a critical look. We have since dropped this reference, and focus on more specific references that focused on FSWs;

Faini D, Munseri P, Sandstrom E, Hanson C, Bakari M. Awareness, willingness and use of HIV pre-exposure prophylaxis among female sex workers living in Dar-es-Salaam, Tanzania. AIDS and Behavior. 2023;27(1):335-43.

Guure C, Afagbedzi S, Torpey K. Willingness to take and ever use of pre-exposure prophylaxis among female sex workers in Ghana. Medicine. 2022;101(5): e28798.

Witte SS, Filippone P, Ssewamala FM, Nabunya P, Bahar OS, Mayo-Wilson LJ, et al. PrEP acceptability and initiation among women engaged in sex work in Uganda: Implications for HIV prevention. EClinicalMedicine. 2022;44.

Nwagbo E, Ekwunife O, Mmeremikwu A, Ojide C. Awareness of and willingness to use pre-exposure prophylaxis to prevent HIV infection among female sex workers in Anambra State, south-eastern Nigeria. African Journal of Clinical.

Witte SS, Filippone P, Ssewamala FM, Nabunya P, Bahar OS, Mayo-Wilson LJ, et al. PrEP acceptability and initiation among women engaged in sex work in Uganda: Implications for HIV prevention. J EClinicalMedicine

5. Lines 9-10: References 7 and 8 evidence PrEP acceptability rates of 91 and >50%. Not clear what data supports the 100% acceptability rates stated here

Response: Thank you for this observation. We have now only focused on data that focused on FSW as per the references in 4 above.

6. The authors should define PrEP persistence, and clarify the distinction made, if intentional between persistence and long-term use. They should also, clarify what is meant by diverse settings.

Response: We apologize for the confusion caused. We have defined PrEP persistence to mean sustained use over time in this study. We also, agree with the reviewer long term use and diverse settings are confusing and have been deleted.

7. The authors should clarify what is meant by everyday practice.

Response: we apologize for not being clear. We had intended to mean PrEP adoption as a prevention intervention by FSWs. However, we have no deleted it altogether.

Methods:

1. Line 15: The authors should clarify the language (s) in which the questionnaire was administered and proficiency of both parties in the language(s).

Response: We appreciate the reviewers for this recommendation. The text below has been provided under data collection (page 6, lines 16-18);

Interviews were conducted by a trained team member fluent in the respondent’s preferred language (English, Luganda or Runyankore).

Results:

1. Lines 15-17; Average number of clients per week may better be described as a behavioral factor than a demographic characteristic.

Response: We thank the reviewer for this suggestion. We also, agree that is another context it could be presented as a behavioral factor. However, we believe for this current article (reality/phenomenon), presenting it under demographics describes the demand for their services.

2. Line 21: Based on the data in Table 1, the percentage of FSWs with primary or secondary education in Mbale is 90% not 85%.

Response: We apologize for the error. We had erroneously indicated 35 instead for 30 for secondary education, this has since been corrected in Table 1.

3. Line 22: The authors should clarify that marital status was a significant difference between the two cities

Response: We thank the reviewers for this observation and suggestion. We have now implemented the recommendation. The following text has been included (page 8, lines 3-5);

Marital status significantly differed between the cities, with most FSWs in Mbale identifying as single (61%) and most in Mbarara describing themselves as separated (54%).

4. Line 4-6 (page7): The authors should include PEP awareness and usage. They should also, consider specifying syphilis testing and HIV rather than generalizing STI/HIV testing.

Response: We are grateful to the reviewers for these suggestions. We have now implemented them. The text below has been included (page, 8 lines, 18-19 & 9-14);

Also, PEP awareness and use were lower in Mbale (awareness: 45%, use: 8%) compared to Mbarara (awareness: 66%, use: 15%)

Page 8, lines 9-14

Compared to Mbale, FSWs in Mbarara reported higher rates of recent syphilis and HIV testing. Lifetime syphilis testing was significantly higher in Mbarara (97%) than in Mbale (61%), with recent testing (within the past three months) also more common in Mbarara (87%) than in Mbale (26%). Although lifetime HIV testing rates were similar between the two sites (Mbarara: 97%, Mbale: 93%), recent HIV testing was more frequent in Mbarara (82%) than in Mbale (52%).

Discussion:

1. Lines 17-18 (page 7): The authors should opt for more precise language than “generally low” to describe what are the principal findings of the study. Additionally, in the abstract, the authors describe PrEP use in this cohort to be “very low,” which is substantially different from how it is described here. The authors should be consistent with the language used to describe these findings.

Response: we are sorry for the choice of language and inconsistence in reporting our findings. We have now revised both in the discussion and abstract to be more precise and consistent. In our interpretation both in the discussion and abstract we are only using low instead of very low. The text below has been included to replace general low (page 9, lines 5-6)

Less than half (42.3%) of the FSWs reported being aware of oral PrEP, with only 9.7% reporting current use.

2. Lines 21-22 (page 7): The authors should specify the “services” to which they refer here.

3.

Response: We are sorry for being unclear. We have since corrected this to refer to PrEP services (page 9).

4. Lines 22-4 (pp. 7-8): The authors should switch the order of these sentences (i.e., “Notably…” before “The differences in…”) to present the relevant findings prior to the speculative clause.

Response: We thank the reviewers for this observation and suggestion. We have now implemented the recommendation (page 9, lines 11- 17)

5. Lines 2-3 (page 8): The authors should provide a reference here for the PrEP demonstration project in Mbarara that is mentioned.

Response: We are sorry for not providing this reference. We have now provided a reference for a PrEP project implemented at Mbarara regional referral hospital targeting women at increased risk of HIV infection

Matthews LT, Atukunda EC, Owembabazi M, Kalyebera KP, Psaros C, Chitneni P, et al. High PrEP uptake and objective longitudinal adherence among HIV-exposed women with personal or partner plans for pregnancy in rural Uganda: A cohort study. PLoS medicine. 2023;20(2):e1004088.

6. Lines 4-5: The authors should refine this sentence (“Despite…”) to be more precise. This study assessed current usage of oral PrEP services among FSWs in two cities in Uganda. The findings, as written here, seem to refer to the current usage of oral PrEP services among the general population across all of Uganda.

Response: We apologize for being unclear. We have revised the sentence and reference to FSWs (page 9, lines 19-20).

7. Lines 5-6: This sentence needs to be reworked. The findings that current usage of oral PrEP services among FSWs was low do not directly “contrast” those of previous studies that evidence high willingness among FSWs to use oral PrEP; rather, they simply highlight that willingness to use (which is different from “intention,” and would make this change as well) and actual usage are two different things.

Response: We are sorry for this confusion. We have now corrected the sentence to acknowledge the fact that willingness to use and actual use are not the same (page 10, lines 3-5)

8. Lines 11-16: The authors should specify what is meant by “key populations” on lines 15-16 i.e., populations at increased risk for HIV? Additionally, the authors should consider including a reference or two that are specific to the population of interest (i.e., FSWs) if possible.

Response: We thank the reviewers for this observation. We have now adopted the term “population at increased risk for HIV” throughout the manuscript. We have also, included references specific to FSWs

Faini D, Munseri P, Sandstrom E, Hanson C, Bakari M. Awareness, willingness and use of HIV pre-exposure prophylaxis among female sex workers living in Dar-es-Salaam, Tanzania. AIDS and Behavior. 2023;27(1):335-43.

Guure C, Afagbedzi S, Torpey K. Willingness to take and ever use of pre-exposure prophylaxis among female sex workers in Ghana. Medicine. 2022;101(5): e28798.

Witte SS, Filippone P, Ssewamala FM, Nabunya P, Bahar OS, Mayo-Wilson LJ, et al. PrEP acceptability and initiation among women engaged in sex work in Uganda: Implications for HIV prevention. EClinicalMedicine. 2022;44.

Nwagbo E, Ekwunife O, Mmeremikwu A, Ojide C. Awareness of and willingness to use pre-exposure prophylaxis to prevent HIV infection among female sex workers in Anambra State, south-eastern Nigeria. African Journal of Clinical.

Witte SS, Filippone P, Ssewamala FM, Nabunya P, Bahar OS, Mayo-Wilson LJ, et al. PrEP acceptability and initiation among women engaged in sex work in Uganda: Implications for HIV prevention. J EClinicalMedicine

9. Lines 17-18: Again, the authors should consider specifying what is meant by “those who need it most” i.e., populations at increased risk for HIV.

Response: We thank the reviewers for noticing this error on our part. We have now replaced this with FSWs (page 10 line 6 & 14)

10. Lines 19-3 (pp. 8-9): The authors should include non-random sampling/ recruitment of survey respondents as a possible limitation to the study.

Response: We thank the reviewers for this suggestion. We have now implemented it (page 10 lines 20-21)

Furthermore, peer referrals, a non-random sampling strategy was used to enrol study respondents.

Conclusions:

1. The authors make sound conclusions and suggest practical next steps that follow from the data presented.

Response: We are grateful to the reviewers for this uplifting comment

Table 2:

2. The authors should con

---

## [Decision Letter · Decision Letter 1]

4 Feb 2025

PONE-D-24-24744R1Associations with HIV Preexposure Prophylaxis Use by Cisgender Female Sex Workers in Two Ugandan CitiesPLOS ONE

Dear Dr. Muhindo,

Thank you for submitting your manuscript to PLOS ONE. After careful consideration, we feel that it has merit but does not fully meet PLOS ONE’s publication criteria as it currently stands. Therefore, we invite you to submit a revised version of the manuscript that addresses the points raised during the review process.

We look forward to receiving your revised manuscript.

Kind regards,

Hamufare Dumisani Mugauri, Ph.D. Medicine and Health Sciences

Academic Editor

PLOS ONE

Reviewers' comments:

Reviewer's Responses to Questions

**Comments to the Author**

1. If the authors have adequately addressed your comments raised in a previous round of review and you feel that this manuscript is now acceptable for publication, you may indicate that here to bypass the “Comments to the Author” section, enter your conflict of interest statement in the “Confidential to Editor” section, and submit your "Accept" recommendation.

Reviewer #3: (No Response)

Reviewer #4: (No Response)

2. Is the manuscript technically sound, and do the data support the conclusions?

Reviewer #3: Partly

Reviewer #4: (No Response)

3. Has the statistical analysis been performed appropriately and rigorously? 

Reviewer #3: Yes

Reviewer #4: Yes

4. Have the authors made all data underlying the findings in their manuscript fully available?

Reviewer #3: No

Reviewer #4: (No Response)

5. Is the manuscript presented in an intelligible fashion and written in standard English?

Reviewer #3: Yes

Reviewer #4: No

6. Review Comments to the Author

Reviewer #3: The authors present the results of an investigator-delivered survey administered to 236 FSW at risk for HIV in Mbale and Mbarara, Uganda in early 2020. The focus of the manuscript was use of oral TDF-based HIV PrEP.

Overall knowledge of HIV PrEP and reported use (see below was low in a group of individuals with heighted risk of HIV acquisition. Data that sheds light into knowledge and use of PrEP in this group is worthy of attention. The differences by location and educational access in the adjusted models are interesting though not surprising.

The main issue with the manuscript is the lack of clarity around the primary outcome. It is not clear if this refers to current daily oral PrEP use, or ever use, or use in the prior 3 months. Please make it clear the time frame for PrEP use. In the abstract the authors describe current use. We are to assume that the regression is based on N=23. This means that the reference cell in the City variable in Table 3=0?

The other important issue is a more thorough explanation of differences by location in the discussion. This is important for readers unfamiliar with Uganda. Why might Mbarara have better awareness? Is it just that the demonstration project had increased awareness and use. What is the role of HIV prevalence, ongoing HIV-focused research that increase community-level awareness of HIV and HIV prevention?

Minor issues to consider.

Including the SAPPH-IRe trial would be useful.

Affiliations: What institution for the first author?

Abstract: Capital 'A' needed line 18.

Intro:

1. Consider deleting 'in less than a decade' - PrEP FDA-approved in 2012. Or do you mean that the role of PrEP for FSW has only been understood for <10 years? The SAPPH-IRe trial was 2014-16.

2. Page 4. Lines 13-15. Are there specific data on role of FSW? The specific data are 'hidden' in the total of the KP numbers. It would be much better here to cite the original ref (Korenromp et al 2024) and give the data on partners of SW - table 2, rather than a composite).

3. Page 4. Lines 16-18 provides better information than lines 13-15. Suggest sing this instead.

4. Page 5. Lines 4-11. This is interesting but does not really relate to the described outcomes which describe knowledge and use. Not persistence. I would suggest shortening this. It gives the impression that this is the focus of your analysis.

Methods:

1. Page 6. Lines 1-2. What does that mean? What was the baseline HIV testing uptake during that time period compared with 2020? Important context. The 2019 AIDS Res Ther showed that 67.4% of FSW took at least 2 HIV tests in the prior 12 months. Which is essentially the same as you report here.

2. Sex work hotspots - page 6. Line 6. Move up the explanation from lines 10-11.

3. Can you explain what expertise barmaids have? This will not be clear to some readers in places where barmaid work is not related to sex work knowledge.

4. Page 6. Lines 21-22. Page 7. Lines 2-3. Per the comment above is this current or ever use? What is the timeframe? I assume from the abstract that it is current use, it would be good to make this very clear.

5. Page 7. Lines 11-15. Were the participants compensated?

Discussion:

1. Page 9. Line 7. is it correct to say 'three times' Range was 2.5-3.1. Maybe 2.5-3x.

2. Page 9. Lines 16-17. This seems somewhat tangential as is HIV-focused. This needs some more explanation if it is to stay in. Do the authors mean that physical location is the determining factor or transportation, educational activities, presence of NGOs etc wrt to HIV services?

3. Page 9. Line 19. Again, clarify the 'when' of PrEP use.

4. Some of the discussion is repetitive. For example, the authors could make more succinct by combining. Page 9. Lines 20-21 with Lines 9-10. Page 10. Lines 5-6. Similar to lines 9-10, 20-21. Could combine all of these. Page 10.. Lines 7-9 are very similar to Page 9. Lines 7-9. Could combine.

5. The authors mention 'willingness' on page 10, lines 3-4. You did not measure willingness in your population, correct? I would make clear that you are assuming willingness in other studies equates to willingness in FSW in Mbale and Mbarara.

Conclusions:

1. Page 11. Lines 5-8. This is essentially repetition of lines 21-22 on page 10.

References:

#9. I cannot find the data referenced in the text when I click on this link. Why are the authors using 2021 Ugandan fact sheet? Are there more recent data available?

#10. Are there no more recent data than 2009?

Tables:

All need a key with abbreviations spelled out.

Table 1. Typology is ambiguous here. Suggest 'sex work venue' or similar.

Escort is a type of SW not a venue.

Table 2.

Ever hear - add 'of PEP'.

0.90 - text says 97.

0.81 - text says 87.

Reviewer #4: The manuscript highlights a critical and timely issue, as female sex workers (FSWs) remain a population of focus for HIV prevention efforts across sub-Saharan Africa (SSA). The gaps in accessing effective prevention services within this population underscore the significance of the study's findings and the contribution it could make to the field.

The authors have commendably addressed most of the concerns raised by the previous reviewers, demonstrating their commitment to improving the manuscript. However, several issues still require attention to enhance the clarity, rigor, and overall impact of the paper.

I recommend the following revisions:

1. The aim and study design:

There are inconsistencies in the study aim, design, and outcome as presented across different sections of the manuscript.

• Abstract: In a quasi-experimental study to assess the effect of peer education and text message reminders on the uptake of regular sexually transmitted infection (STI) and HIV testing, we administered a survey questionnaire to 236 HIV-negative FSW in the cities of Mbale and Mbarara from January to March 2020

• Introduction: Page 4, lines 19-21 This study aimed to assess PrEP use and associated factors among FSWs in two urban settings in Uganda, providing valuable insights into PrEP program implementation in everyday practice

• Discussion: Page 7, lines 17-18 This cross-sectional study investigated oral PrEP awareness and use among FSWs in Uganda, revealing a generally low awareness and use

The authors should review these and ensure consistency throughout the manuscript.

2. Key Terminology

The authors have indicated that the primary outcome is PrEP use, defined as self-reported use of daily oral pills. However, it is unclear whether they meant use at the time of the interview, or for a particular period. In the discussion, the phrase ‘current use’ is used repeatedly. Providing a clearer definition will ensure consistent understanding among diverse readers.

3. Study setting:

While the authors have addressed the previous reviewer’s question regarding the selection of Mbale and Mbarara for this study, the explanation provided in the methods section remains unclear. Please review this section to ensure the rationale is clearly articulated.

4. Discussion:

While the authors discuss the main findings, the writing needs to be tightened for more coherence and clarity including appropriate citations. e.g the discussion of the correlation between education level and PrEP uptake. In addition, the authors have made a comparison between their findings and those of studies evaluating willingness to use PrEP, while there are a lot of publications on PrEP uptake among FSW in similar settings.

7. PLOS authors have the option to publish the peer review history of their article (what does this mean? ). If published, this will include your full peer review and any attached files.

**Do you want your identity to be public for this peer review?** For information about this choice, including consent withdrawal, please see our Privacy Policy .

Reviewer #3: No

Reviewer #4: No

---

## [Author Response · Author response to Decision Letter 1]

10 Feb 2025

Dr. Hamufare D Mugauri

Academic Editor, Plos One

February 8th, 2025

Dear Dr. Hamufare,

Regarding PONE-D-24-24744

We are very grateful for the reviewer comments provided by the reviewers of our manuscript: “Associations with HIV Preexposure Prophylaxis Use by Cisgender Female Sex Workers in Two Ugandan Cities ". Reviewer comments are encouraging, and the reviewers appear to share our view that study findings are of public health importance. The suggestions offered by the reviewers have been greatly helpful. Please find below our point-by-point responses in italics, with new text in bold font:

Reviewer Comments

Reviewer 3

1. The authors present the results of an investigator-delivered survey administered to 236 FSW at risk for HIV in Mbale and Mbarara, Uganda in early 2020. The focus of the manuscript was use of oral TDF-based HIV PrEP. Overall knowledge of HIV PrEP and reported use (see below was low in a group of individuals with heighted risk of HIV acquisition. Data that sheds light into knowledge and use of PrEP in this group is worthy of attention. The differences by location and educational access in the adjusted models are interesting though not surprising.

Response: We thank the reviewers for this encouraging overall comment regarding our work.

2. The main issue with the manuscript is the lack of clarity around the primary outcome. It is not clear if this refers to current daily oral PrEP use, or ever use, or use in the prior 3 months. Please make it clear the time frame for PrEP use. In the abstract the authors describe current use.

Response: We thank the reviewer for this observation. We are sorry for being unclear about the primary outcome. However, our primary outcome was current use/self-report of oral PrEP use at the time of data collection. Respondents were asked if they were currently taking daily pills to prevent HIV acquisitions. The text below has been added (page 6, line 17 & 20).

The primary outcome was current PrEP use, defined as self-reported use of daily oral pills to prevent HIV acquisition at the time of the interview.

3. We are to assume that the regression is based on N=23. This means that the reference cell in the City variable in Table 3=0?

Response: We thank the reviewers for this observation. True our regression is based current use, and therefore in a sample of 236, only 23 reported use at the time of data collection in Mbarara, we therefore thought our approach to model this data (treated as count data) using a modified Poisson regression, with the ability to account for any potential overdispersion or correlation in the data, providing more reliable inference was ok. However, we are open to suggestions to improve this.

4. The other important issue is a more thorough explanation of differences by location in the discussion. This is important for readers unfamiliar with Uganda. Why might Mbarara have better awareness? Is it just that the demonstration project had increased awareness and use. What is the role of HIV prevalence, ongoing HIV-focused research that increase community-level awareness of HIV and HIV prevention.

Response: We thank the reviewers for this recommendation. We have improved the explanation to capture the role of HIV prevalence, ongoing HIV focused research. The text below has been included (page 9, lines 9-17)

Our finding that FSWs in Mbarara were twice as likely to be aware of PrEP and report current use compared to those in Mbale highlights limited awareness and uptake in certain Ugandan urban locations. This disparity may be attributed to Mbarara's high adult HIV prevalence (14.4% vs 5.4% in Mbale), which has likely fueled ongoing HIV-focused research and increased community-level awareness of HIV prevention. Furthermore, the presence of a PrEP demonstration project targeting women at increased risk of HIV acquisition in Mbarara may also contribute to the observed difference

5. Including the SAPPH-IRe trial would be useful

Response: We thank the reviewers for this recommendation. We have reviewed the trial found it very important as it evaluated a dedicated programme/support for female sex workers in Zimbabwe accessing and adhering to antiretrovirals for treatment and prevention. From the article I reviewed, most of the FSWs in the trial were HIV positive, and with the intervention adherence and viral suppression was improved, I would have loved to include it but given that in this study, our focus was mainly to assess current PrEP use among HIV-naïve FSWs I wasn’t sure where to include it.

6. Affiliations: What institution for the first author?

Response: We apologize for not providing complete information on the institutional affiliations of the first author. This has been included. The following text has been included (title page)

1Department of Nursing Makerere University College of Health Sciences

7. Abstract: Capital 'A' needed line 18

Response: We thank the reviewers for noticing this error. We have since implemented the recommendation on page 2, line 18

Introduction

1. Consider deleting 'in less than a decade' - PrEP FDA-approved in 2012. Or do you mean that the role of PrEP for FSW has only been understood for <10 years? The SAPPH-IRe trial was 2014-16.

Response: We thank the reviewer for this recommendation. We have since implemented the recommendation on page 4, line 2

2. Page 4. Lines 13-15. Are there specific data on role of FSW? The specific data are 'hidden' in the total of the KP numbers. It would be much better here to cite the original ref (Korenromp et al 2024) and give the data on partners of SW - table 2, rather than a composite).

Response: We thank the reviewers for this suggestion. We have since implemented it in the introduction (page 4, line 13). The text below referenced to Korenromp et al 2024 has been included.

In 2022, sex workers accounted for an estimated 5.2-15% of new HIV infections in Sub-Saharan Africa

3. Page 4. Lines 16-18 provides better information than lines 13-15. Suggest sing this instead

Response; We thank the reviewer for this encouraging comment. We have revised lines 13-15 to provide information that focuses on sex workers on page 4, line 13-17

4. Page 5. Lines 4-11. This is interesting but does not really relate to the described outcomes which describe knowledge and use. Not persistence. I would suggest shortening this. It gives the impression that this is the focus of your analysis.

Response: We thank the reviewer for this observation. We have implemented the suggestion by deleting persistence and focus on use (page 5, lines 4-9)

Methods:

1. Page 6. Lines 1-2. What does that mean? What was the baseline HIV testing uptake during that time period compared with 2020? Important context. The 2019 AIDS Res Ther showed that 67.4% of FSW took at least 2 HIV tests in the prior 12 months. Which is essentially the same as you report here.

Response: We sorry if our statement regarding the rationale to select Mbarara and Mbale was unclear. While our BMC Health Serv Res 21, 436 (2021). https://doi.org/10.1186/s12913-021-06461-w article gives a better picture of the changes between 2018 and 2020, we have removed the statement and indicated that Mbale and Mbarara served as control and intervention sites for our quasi-experiment, and as such were purposely selected for the PrEP use study (page 5, lines 17-19). The text below has been included

Mbarara and Mbale served as the intervention and control sites, respectively, for our quasi-experimental study, and were purposively selected for the PrEP use survey.

2. Sex work hotspots - page 6. Line 6. Move up the explanation from lines

10-11.

Response: we thank the reviewers for this recommendation. We have implemented as suggested (pages 5, line 22, and page 6, lines 1-10)

3. Can you explain what expertise barmaids have? This will not be clear to some readers in places where barmaid work is not related to sex work knowledge.

Response: We thank the reviewers for this observation. We have included the text below to implement this recommendation (page 6, line 4)

In Uganda, barmaids may engage in sex work or facilitate connections between sex workers and clients

4. Page 6. Lines 21-22. Page 7. Lines 2-3. Per the comment above is this current or ever use? What is the timeframe? I assume from the abstract that it is current use, it would be good to make this very clear.

Response: We apologize for being unclear. We have since consistently referred to current use, defined as self-reported use of daily oral pre-exposure prophylaxis (PrEP) pills to prevent HIV acquisition at the time of the interview throughout the article (page 6, lines 17 & 20)

5. Page 7. Lines 11-15. Were the participants compensated?

Response: We are sorry for not stating this. However, all participants were compensated for their time. The text below has been included (page 7, lines 11-12)

All respondents provided written informed consent in English or their local language, and were compensated for their time.

Discussion:

1. Page 9. Line 7. is it correct to say 'three times' Range was 2.5-3.1? Maybe 2.5-3x.

Response: We thank the reviewers for this suggestion. We have since implemented the recommendation (page 9, line 5)

2. Page 9. Lines 16-17. This seems somewhat tangential as is HIV-focused. This needs some more explanation if it is to stay in. Do the authors mean that physical location is the determining factor or transportation, educational activities, presence of NGOs etc wrt to HIV services?

Response: We apologize for this error on our part. We have since revised to highlight the role of ongoing research and programs. The text below has been included (page 9, lines 10-18)

Our finding that FSWs in Mbarara were twice as likely to be aware of PrEP and report current use compared to those in Mbale highlights limited awareness and uptake in certain Ugandan urban locations. This disparity may be attributed to Mbarara's high adult HIV prevalence (14.4% vs 5.4% in Mbale), which has likely fueled ongoing HIV-focused research and increased community-level awareness of HIV prevention. Furthermore, the presence of a PrEP demonstration project targeting women at increased risk of HIV acquisition in Mbarara may also contribute to the observed difference

3. Page 9. Line 19. Again, clarify the 'when' of PrEP use.

Response: We sorry, and thank the reviewers for noticing this. We have since consistently used current use in the manuscript.

4. Some of the discussion is repetitive. For example, the authors could make more succinct by combining. Page 9. Lines 20-21 with Lines 9-10. Page 10. Lines 5-6. Similar to lines 9-10, 20-21. Could combine all of these. Page 10. Lines 7-9 are very similar to Page 9. Lines 7-9. Could combine.

Response; We are sorry for this error. We have implemented the recommended and now the discussion goes as below (page 9, lines 1-23

This cross-sectional study investigated current oral PrEP use among FSWs in two Ugandan cities. Less than half (42.3%) of the FSWs reported being aware of oral PrEP, with only 9.7% reporting current use. However, awareness and use significantly varied by location. Notably, FSWs who had attained secondary or tertiary education were 2.5-3 times more likely to report current use, consistent with previous studies conducted in SSA (29-31). These studies have consistently shown a positive correlation between education level and PrEP usage, highlighting the need for implementation research on strategies to promote PrEP use among FSWs with lower education levels.

Our finding that FSWs in Mbarara were twice as likely to be aware of PrEP and report current use compared to those in Mbale highlights limited awareness and uptake in certain Ugandan urban locations. This disparity may be attributed to Mbarara's high adult HIV prevalence (14.4% vs 5.4% in Mbale) (10), which has likely fueled ongoing HIV-focused research and increased community-level awareness of HIV prevention (33, 34). Furthermore, the presence of a PrEP demonstration project targeting women at increased risk of HIV acquisition in Mbarara may also contribute to the observed difference (34). Targeted educational campaigns to FSWs regarding the benefit and availability of oral PrEP services are still needed.

Regional variations in PrEP usage have been observed among FSWs in East Africa. While our finding of low current PrEP use (9.7%) is comparable to a Kenyan study (24%) (29), a recent Tanzanian study reported notably higher use (97%) (30). These disparities likely stem from contextual differences in service organization, provider characteristics, and recipient factors. Existing literature cites various barriers to non-oral PrEP use among FSWs, including limited access, lack of awareness, negative perceptions, stigma, and pill burden (29, 38, 39). Despite these insights, targeted interventions are still necessary to improve PrEP uptake and usage.

This cross-sectional study, conducted prior to the first COVID-19 lockdown in Uganda, provides a snapshot of PrEP use among FSWs in Mbale and Mbarara. However, the findings may not capture trends in PrEP use over time. Additionally, self-reported PrEP use may be subject to social desirability bias. Moreover, the study's sample was limited to two cities and may not be generalizable to all FSW communities in Uganda. Furthermore, peer referrals, a non-random sampling strategy was used to enroll study respondents. Despite these limitations, the results highlight the need for targeted educational campaigns and implementation studies to increase PrEP uptake among FSWs, particularly in areas with low awareness and usage beyond demonstration projects.

Conclusions

The low PrEP usage in this FSW cohort underscores the need for implementation studies that evaluate strategies to improve uptake, particularly among those with lower levels of education.

5. The authors mention 'willingness' on page 10, lines 3-4. You did not measure willingness in your population, correct? I would make clear that you are assuming willingness in other studies equates to willingness in FSW in Mbale and Mbarara.

Response: We are sorry for this error. We have now referenced to only use, page 9, lines 19-23, page10, lines 1-3).

Conclusions:

1. Page 11. Lines 5-8. This is essentially repetition of lines 21-22 on page 10.

References: we are sorry. We have amended as follows (page 10, line 15)

The low PrEP usage in this FSW cohort underscores the need for implementation studies that evaluate strategies to improve uptake, particularly among those with lower levels of education

#9. I cannot find the data referenced in the text when I click on this link. Why are the authors using 2021 Ugandan fact sheet? Are there more recent data available?

Response: We are sorry that the provided link is unable to open. We have provided a better link; Uganda AIDS Commission. HIV Factsheet 2023. FACTS ON HIV AND AIDS IN UGANDA 2023 (Based on Data Ending 31st December 2022) [accessed 2025 February] Available from; https://uac.go.ug/index.php 2023. The most recent data is 2024 but doesn’t provide data on FSWs. We have however, used the data of 2022 instead of 2021.

#10. Are there no more recent data than 2009?

Response: Sorry, we used this study, because the 2009 modes of transmission study estimated the contribution of sex work to new HIV infections. While some articles (Witte, Susan S., et al. "PrEP acceptability and initiation among women engaged in sex work in Uganda: Implications for HIV prevention." EClinicalMedicine 44 (2022)), have quoted Uganda AIDS commission fact sheets 2020, review of this fact sheet doesn’t show any information related to the contribution of sex work to new HIV infections.

Tables:

All need a key with abbreviations spelled out.

Response: we thank the reviewers for this suggestion. We have implemented the recommendation (pages 19, line2; page 20, line 2; page 21, line 2)

Table 1. Typology is ambiguous here. Suggest 'sex work venue' or similar.

Response: We thank the reviewer for this suggestion.

Escort is a type of SW not a venue.

Response: We thank the reviewers for noticing this error. We have revised.

Table 2.

Ever hear - add 'of PEP'.

Response: Thank you for noticing this error (page 20, line 2)

0.90 - text says 97.

0.81 - text

---

## [Editor Report · Decision Letter 2]

13 Feb 2025

Associations with HIV Preexposure Prophylaxis Use by Cisgender Female Sex Workers in Two Ugandan Cities

PONE-D-24-24744R2

Dear Dr. Muhindo,

We’re pleased to inform you that your manuscript has been judged scientifically suitable for publication and will be formally accepted for publication once it meets all outstanding technical requirements.

Kind regards,

Hamufare Mugauri, Ph.D. Medicine and Health Sciences

Academic Editor

PLOS ONE

---

## [Editor Report · Acceptance letter]

PONE-D-24-24744R2

PLOS ONE

Dear Dr. Muhindo,

I'm pleased to inform you that your manuscript has been deemed suitable for publication in PLOS ONE. Congratulations! Your manuscript is now being handed over to our production team.

Kind regards,

on behalf of

Mr Hamufare Mugauri

Academic Editor

PLOS ONE